# A User-Driven Approach to Prosthetic Upper Limb Development in Korea

**DOI:** 10.3390/healthcare9070839

**Published:** 2021-07-02

**Authors:** Naan Ju, Kyu-Hye Lee, Myoung-Ok Kim, Youngjin Choi

**Affiliations:** 1Research Institute of Industrial Science, Hanyang University, Seoul 04763, Korea; naan_ju@hanyang.ac.kr; 2Human-Tech Convergence Program, Department of Clothing and Textiles, Hanyang University, Seoul 04763, Korea; khlee@hanyang.ac.kr; 3College of Design, Architecture, Art and Planning, University of Cincinnati, Cincinnati, OH 45221, USA; 4Department of Electrical and Electronic Engineering, Hanyang University, Ansan 15588, Korea

**Keywords:** prosthetic upper limb, user satisfaction, user-centered prosthetic upper limb, upper limb amputee, user’s needs, user’s requirements, assistive technology, technology development

## Abstract

Despite recent significant advances in technology and medicine, the number of patients who undergo amputation of body parts for various reasons continues to increase. Assistive devices such as prosthetic arms can enable limited activities in upper limb amputees and improve their quality of life. This study aims to help in the development of user-centered prosthetics by identifying user requirements and key considerations during selection of prosthetics. This study conducted a questionnaire survey after obtaining prior consent for persons with disabilities with upper limb amputation who visited orthosis companies, rehabilitation centers for the disabled, veteran’s hospitals, and labor welfare corporations. A modified questionnaire was conducted to upper limb prosthetic users and results were analysed using descriptive statistics and *t*-test. Results of the study showed that the main reasons for discontinuing the use of prosthetics were discomfort (discomfort in wear, weight, and difficulty of detachment) and complaints regarding design and function. Regardless of the prosthesis type, the color and design of the prosthesis were key considerations in prosthesis choices. Respondents indicated that they needed various prostheses designed according to the purpose and situation, such as for sports like golf and cycling as well as everyday use. Most of the respondents answered that buttoning shirts, tying knots, and using chopsticks were challenging or impossible to do on their own. Based on the results of this study, the quality of life of upper limb amputees can be improved if a prosthetic arm with various functions that can satisfy both the user’s needs and wants is developed.

## 1. Introduction

Despite remarkable advances in technology and medicine in recent years, the number of patients who undergo amputation due to congenital surgical diseases, various accidents, complications, industrial accidents, etc. is constantly increasing. According to the Korean Ministry of Health and Welfare, disability due to amputation is the most common type of disability among all 15 types of disabilities. The number of patients undergoing an amputation is estimated to be 171,950, and 81.2% of them have amputation disabilities [1]. As compared with lower limb amputation, the incidence of upper limb amputation is relatively low [1], however, upper limb amputation; results in extensive loss of function, such as the loss of the ability to perform almost all daily activities.

The limitation of daily life activities due to disability has a significant correlation with depression of the disabled [2]. Social isolation is a serious consequence of amputation because amputees often significantly decrease social activities and avoid meeting new people [3]. Therefore, to lower depression and relieve social isolation in people with disabilities, the development of assistive devices that enable physical activity, can play a role. Beekman and Axtell [4] argued that amputees believe that assistive devices not only enable limited activities due to disability, but also improve the quality of life, thereby lowering social isolation. However, despite the benefits of using these assistive devices, many amputees with severe upper limb trauma do not use prosthetics for reasons of discomfort, pain, or unusefulness [5,6,7]. In a survey on the status of the disabled in Korea, many people with disabilities responded that they did not purchase assistive devices although they needed them because of the cost, the lack of knowledge about prostheses, and ineffectiveness [1].

In recent years, improving the physical function of people with disabilities and helping them return to daily life activities has emerged as a global concern. However, recent research has focused on the development of new prosthetics utilizing various functions and the latest technologies, investigations of satisfaction with the prostheses and their rate of use have been limited [8,9,10]. According to Biddiss et al. personal and contextual factors play a critical role in prosthesis acceptance [11]. In this study, we examined the status of Korean amputees’ use of prosthetic hands and evaluated their level of satisfaction with the existing prosthetic hands. This study also aimed to identify user requirements and key considerations while selecting a prosthetic hand. This attention should help develop user-centered prosthetic devices that satisfy both the needs and wants of patients suffering from significant psychological and social losses due to upper limb amputation.

## 2. Literature Review

### 2.1. Types and Functions of the Prosthetic Hand

Through the use of hands, a person interacts directly with the external environment and loss of hand use has a direct impact on life situations. A prosthetic hand is a representative assistive device for people who underwent an upper limb amputation and compensates for hand function. In general, prosthetic hands refer to cosmetic hands designed only for esthetic purposes. Furthermore, body-powered hands have been developed to perform grasping movements by using the shoulder movements. However, these body-driven prostheses require movement of the shoulder or upper body for operation, making them uncomfortable to use for long periods [12].

The electric-powered prosthetic hand, which was proposed to solve the problems of body-driven prostheses, was first introduced in 1954 announced as an industrial robot with versatility for repetitive work. The battery for the electric-powered prosthetic hand, which is shaped like a regular hand, was used to generate motive power. The motor-driven prosthetic arm is used as a control signal for hand movements such as grip or wrist rotation by recognizing the myoelectric signal detected from the residual muscle of the cut area, leading to the proposal of a myoelectric hand prosthesis. This muscle anterior prosthesis controller compares the mean absolute value of the muscle potential signal generated by the voluntary contraction of the user’s residual muscle with a preset threshold value to determine the user’s intention [13]. The most representative commercialized myoelectric prosthesis is the System ElectroHand (OttoBock Co., Duderstadt, Germany) and Utah ProControl (Utah Arm Co., Salt Lake City, UT, USA) [14]. Currently, in the United States and Europe, many myoelectric prostheses driven by myoelectric sensors are being studied; in particular, multi-degrees of freedom and lightweight prosthetics are being developed for upper limb amputees who were injured during the Korean war.

However, many people with upper limb amputation disabilities generally use cosmetic prostheses, and according to a study by Jang et al. [15], 80.2% of the users use cosmetic prostheses whereas only 0.37% of users utilize electric prostheses. In this study, the following research questions were created to help develop a prosthesis suitable for users by clarifying the status of prosthetics use by people who underwent upper limb amputations in Korea.

### 2.2. Factors Contributing to the Demand and Rejection of Prosthesis

According to the Ministry of Health, Welfare & Korea Institute for Health and Social Affairs [1] survey on the status of people with disabilities in Korea, many people with disabilities need assistive devices but do not purchase them. The most common reasons for not purchasing assistive devices were the cost, lack of knowledge about assistive devices, and aesthetic reasons. In other words, economic difficulties are a major factor preventing the purchase of necessary assistive devices for the disabled. Furthermore, complaints related to the supply and delivery system of assistive devices for the disabled, such as a lack of awareness regarding suitable assistive devices and the location of purchase, are still not resolved. On the other hand, many people with disabilities possessed an assistive device but did not use it. These people did not use the assistive device because of lack of assistive devices suitable for individuals, inability to wear due to changes in the body shape, or difficulty in use. Among people with an amputated upper limb, only 0.9% of people with disabilities use a prosthetic upper limb and most people who use prosthetic products utilize a low-quality, low-cost, heavy-prosthetic arm. Unfortunately, for some low-cost orthoses that have not undergone proper evaluation for stability and performance, there is a risk of major accidents due to malfunction.

According to Millstein et al. [16], factors that influence the use of prostheses include convenience, functionality, appearance (design), and the degree of cutting. The weight, cost, lifelike design, and function were selected as important factors considered by patients who underwent upper limb amputation [17]. Young women particularly tend to prefer an esthetic prosthesis rather than a prosthetic with excellent function [13], indicating that design and function are important factors while selecting a prosthesis. On the other hand, according to previous studies on wearing and rejecting a prosthesis, Burroughs and Brook [18] reported that the rejection rate was 60% among patients who underwent upper limb amputation. Davidson [19] also concluded that 56% of upper limb amputees did not wear a prosthesis or only wore it occasionally. According to a study conducted by Jang et al. [15] on patients who underwent upper limb amputation in Korea, 18.3% of the respondents said they rarely wear a prosthetic arm, and only 30% were satisfied with the prosthetic arm they used. The reasons for refusing to wear a prosthesis include discomfort, pain, unsatisfactory function, a need for training to use the prosthesis, difficulty in controlling the prosthesis, and the unusefulness [14,16,20]. This means that although many institutions around the world are developing prostheses that can complement the function of the hand, patients are still reluctant to utilize prostheses due to their cost and appearances.

In general, a successful product design requires continuous collaboration and interaction between designers and end-users. In the field of assistive technologies, in which the prosthesis is a close extension of the body, it is important to resolve the user’s discomfort, meet user’s various requirements, and maintain a high level of quality.

### 2.3. Activities of Daily Living (ADL)

Disabled people have limited ADL, such as bathing, dressing, and toilet use. Furthermore, instrumental activities of daily living (IADL), such as housework, hobbies, and social activities are also subject to restrictions. The daily life performance test is used to evaluate people’s ability to move the body for tasks required for living, and the most used test method measures the ability to perform daily life movements. ADL is categorized into physical ADL, which includes the most basic physical activities performed in daily life, IADL, which comprise activities that are required for living such as cleaning, washing, using transportation, preparing meals, and using the phone [21]. Such limitations, and the inability to perform ADL and IADL increase depression and further deteriorate the quality of life of persons with disabilities [22,23]. The level of daily life movements and instrumental daily life movements led to a lower degree of depression [22,24,25,26,27,28]. Thus disabled individuals continuously requires the help of others in performing daily life movements, and if people cannot independently perform daily life movements for a long time, they may easily get depressed [29]. This can, in turn, act as factors that negatively affect rehabilitation and treatment [29].

Consequently, active physical activities of persons with disabilities expand their talents, prove their abilities, eventually improving their quality of life [30]. Winnick [31] argued that the physically or psychologically limited range of activities and variables such as environment or dependence limit participation, resulting in poor physical activity participation by the disabled. It has been reported that the physical activity participation rate of people with physical disabilities is lower than that of non-disabled individuals because most of the physical activities performed by people with disabilities are passively performed [32,33]. Therefore, this study raised the following research questions to help develop user-centered prostheses that can assist the active physical activity of the disabled with upper limb amputation and assist them in daily life movements:

**Research** **Question** **1.**
*What is the actual use of prostheses in upper limb amputees?*


**Research** **Question** **2.**
*What are the factors influencing the prosthesis rejection in upper limb amputees?*


**Research** **Question** **3.**
*What are the factors that upper limb amputees consider when choosing a prosthetic arm?*


**Research** **Question** **4.**
*What is the degree of daily life movements of disabled people with upper limb amputations?*


**Research** **Question** **5.**
*What are some daily life movements that are difficult to perform by disabled individuals with prosthetic upper limbs?*


## 3. Method

### 3.1. Data Collection

This study conducted a questionnaire for upper limb amputees and was classified as a human research survey according to the Bioethics Act. The institutional review board (IRB) approved the research plan after deliberation. After receiving approval (IRB approval number: HY-16-03-12), research was conducted according to the standard procedure. Research participants were notified that the collected contents would not be used for purposes other than research, and the confidentiality and anonymity of the contents were explained in detail along with an assurance that all research-related data would be destroyed after 3 years.

The participants of this study were athletes who underwent upper limb amputation who visited Park-Euiji, a company specializing in assistive devices; the rehabilitation center for the disabled in Icheon; and the Hanyang University Seoul Hospital. Furthermore, the Central Veterans Hospital, and Labor Welfare Corporation Incheon Hospital also approved this study. Researchers who were fully educated about the research objectives and data from questionnaire items collected data through in-person contact, provided consent to disabled persons with upper limb amputations to participate in this study, and confirmed their participation by signing and stamping. The questionnaire was distributed to 200 individuals from 1 June 2016 to 31 August 2016, and 98 individuals agreed to participate in the survey. Finally, the results of 59 responses were used for the analysis, excluding unscrupulous responses. Descriptive statistics and *t*-test were conducted using SPSS Statistics version 26 (IBM, Armonk, NY, USA).

### 3.2. Questionnaire Composition

The questionnaire items comprised questions regarding the general health status of the respondents, whether they use prosthetics, their satisfaction with the assistive devices, their ability to perform daily activities, and factors that they considered when selecting a prosthesis. To measure the use and satisfaction of the prosthesis, the items used in previous studies [15,20] were used. The degree of satisfaction with each item was measured using a 5-points Likert scale (1: very dissatisfied-5: very satisfied). To measure the daily life performance of upper limb amputees, Barthel activities of daily living developed by Mahoney and Barthel [34], the Korean ADL measurement tool, and the Korean IADL measurement tool [35] were modified, supplemented to match the purposes of the study, and used as part of the questionnaire. Specifically, items evaluating behaviors related to movement and urinary control that were not significantly related to prosthetic use were not included. The daily life performance questionnaire is widely used in various fields such as physiology, medicine, and geriatrics, and its contents include eating, bathing, washing, urinating, defecating, dressing, and toilet use. Usually, daily life performance is obtained by a doctor, nurse, or clinical psychologist in the form of questionnaires, interviews, or observations through patients or guardians. However, because it is difficult to accurately observe each life behavior within a limited time, a questionnaire was constructed so that patients and guardians could respond. There was also a space for comments on this in the questionnaire.

For the study, researchers first constructed a draft questionnaire by referring to previous studies. Afterward, this questionnaire was evaluated by a group of experts, including professors of rehabilitation medicine, prosthetic developers, and employees of prosthetic manufacturers, and was finally used for research after reviewing two persons with disabilities with upper limb amputation using prosthetic prostheses. As a result of the review, the overall length of the questionnaire was reduced in consideration of the condition of the respondent, and only those who wanted to respond were allowed to respond to sensitive responses such as educational background and marital status. In the case of the use of the prosthesis, questions were added to answer the reasons why they did not continue to use the prosthesis because the pilot study had subjects who stopped using the prosthesis. For the question about the time it takes for the prosthesis to be convenient to use, the response ‘It is not convenient to use the prosthesis’ was added to reflect the opinion that the prosthesis is not convenient to use even after a long period of time. The revised questionnaire was conducted after receiving final feedback from experts and IRB members that it was appropriate for conducting the study.

## 4. Results and Discussion

### 4.1. Characteristics of the Study Subjects

Among the study subjects, 88.1% were males and 11.9% were females, and the age of the subjects ranged from 10’s years to the 80’s with an average age of 59.04 years, of which subjects in their 40–70’s accounted for 82.1%. Most of the causes of disability were accidents (89.8%), and there were complications and amputations due to war injuries. On the other hand, an amputation caused by a congenital reason was only 1.7% of the total amputations, indicating that most of the amputations occurred due to acquired reasons. The average age at which the respondents underwent amputation was 29.67 years, and the respondents mostly spent an average of 30.36 years after amputation. The duration of prosthesis use ranged from ≤1 month to a maximum of 58 years, and the average duration of use was 21.03 years. Of the total respondents, 16.9% (10 patients) had bilateral amputations and 83.1% (49 patients) had one amputation. Transradial amputation was the most common (32.4% of the respondents), followed by wrist disarticulation (19.1%), partial hand and finger amputation (17.7%), and trans-humeral amputation (16.2%), shoulder disarticulation (8.8%), and elbow disarticulation (including bilateral amputation) (5.9%). (Table 1).

### 4.2. Current Status of Prosthesis Use and Reasons for Collection

According to the results, 93.2% of the respondents said they had used a prosthetic hand; however, 6.8% said they had never used one (Table 2).

Respondents with experience using a prosthesis were asked to select all the prostheses they have used by far and all the prostheses they were currently using. As a result, cosmetic prostheses (70.7%) were the most used protheses followed by semi-automatic prostheses (hooks, 20.7%), semi-automatic prostheses (hands, 5.2%), and electronic prostheses (3.4%). Disabled Koreans with amputation of the upper limb were found to use cosmetic prostheses the most.

When asked to describe the color of the prosthesis utilized, a prosthesis that matched an individual’s skin color was 96.2% (*n* = 50) of all the colors used, indicating that most people with disabilities of upper limb amputation in Korea prefer an inconspicuous color. Factors that prevented people from using a prosthesis included “weight (23.8%),” “functional dissatisfaction (19.0%),” ”design dissatisfaction (14.3%),” “wearing dissatisfaction (14.3%),” “not necessity (9.5%),” “high cost (9.5%),” and “inconvenience of detaching the device (9.5%)” (Table 2).

As shown in Table 3, 10.9% of the respondents answered that they had experience using a prosthesis; however, they currently did not use one. Reasons for not using a prostheses included “weight (15.6%)”, “discomfort on wearing (15.6%),” “functional dissatisfaction (12.5%),” “inconvenience in removing the device (12.5%),” “no necessity (9.3%),” “expensive price (6.3%),” “medical reasons (6.3%),” such as pain/skin trouble, and so on.

As for the rehabilitation training for prosthetic wear (Table 4), 70.9% answered that they had never received rehabilitation training, and only 29.1% had received rehabilitation training. Of the participants, 87.5% received rehabilitation in hospitals, and 12.5% received rehabilitation from the prosthesis manufacturers.

This is consistent with previous studies that say that the reasons for rejecting prostheses use of including discomfort, difficulty in controlling the prostheses, pain, unsatisfactory function, and the need for training to use a prosthesis [11,15,20]. In a situation wherein a person feels dissatisfied with the design, it is challenging to obtain psychological satisfaction because the cosmetic prosthesis cannot match the shape or color of an actual hand. Participants responded that they usually put their hands in their pockets or wear gloves when they used prosthesis, indicating a high sense of dissatisfaction with the esthetic of the prosthesis.

Prosthetic users used a prosthesis for an average of 12.06 h a day, ranging from 2h to 24 h. As a result of responding to the time it took for the prosthesis to be worn and getting accustomed to it, 16.3% of the participants responded ≤6 months, 10.2% responded 6 months-1 year, 18.4% responded 1–3 years, and 6.1% responded ≥3 years. Of the participants, 49.0% (*n* = 24) answered that they were still not comfortable using the prosthesis due to its weight and uncomfortable fit (Table 5).

### 4.3. Prosthesis Satisfaction and Considerations When Choosing a Prosthesis

Color, design, price, weight, function, fit, and ease of detachment were selected as the key considerations while evaluating prosthesis satisfaction according to previous studies. The results are shown in Table 6. Respondents were satisfied with the key considerations of the prosthesis they were currently using in the following order: design (3.57), color (3.42), ease of detachment (3.37), fit (3.2), price (3.1), function (3.02), and weight (3.02). Unlike the data from the Ministry for Health and Welfare and the Korea Institute for Health and Social Affairs [1], the reason that the result of being satisfied with the price was above average was that, unlike general physically disabled people, most of the respondents were injured from industrial accidents or military service, and therefore, received subsidies from the government or companies. However, many respondents found the prosthetics to be expensive considering the quality of the product, and the subsidies were insufficient to change or repair the prosthesis frequently. According to the 2017 Survey on the Disabled in Korea, 36.8% of the disabled responded that they had received various support from outside when purchasing an assistive device. As for the form of support when purchasing an assistive device for the disabled, 94.6% of all respondents received support for part or all the purchase cost, and only 5% of respondents received support in the form of free or paid rental [1]. Through this, various public and private support related to the purchase of assistive devices for the disabled in Korea is focused on the purchase cost support, and support through rental is very insignificant. Considering that leasing-type support accounts for a considerable proportion in major advanced foreign countries such as the US, it is necessary to diversify the form of support in the future. The survey revealed that health insurance and medical benefit insurance programs were the most supported institutions (projects) for purchasing assistive devices [1]. Health insurance in Korea is provided by the National Health Insurance (NHI) program and the coverage of NHI was 97.2% of the population in 2018 [36]. The NHI covers a set benefits package that focuses on curative care, including diagnosis, treatment, traditional medical care, emergency care, pharmaceuticals and dental care [37].

The main considerations (in order to importance) when selecting a cosmetic prosthesis showed that ease of attachment and detachment (4.63), weight (4.56), fit (4.51), design (4.46), and color (4.38) were relatively important (Table 7). However, price (4.27) and function (4.26) were relatively less important because cosmetic prostheses are used for cosmetic, rather than functional purposes. Next, the primary considerations when selecting an electronic prosthesis included function (4.74), weight (4.72), fit (4.71), ease of detachment (4.66), color (4.35), design (4.43), and price (4.43). The reason that the price is relatively less important when choosing an expensive electronic prosthesis as compared to a cosmetic prosthesis is that, as mentioned above, most of the respondents suffered physical disabilities due to industrial accidents, war, or military accidents, and therefore, received subsidies from the government or companies. On the other hand, a *t*-test was conducted to determine whether there is a difference in considerations when selecting a cosmetic prosthesis versus an electronic one, it appears that there is only a difference in function. Therefore, we conclude that people who underwent upper-body amputation tend to emphasize the functional aspect of the product when selecting an electronic prosthesis rather than an esthetic one (Table 7).

### 4.4. ADL

As a result of measuring the ADL of those with an amputation of the upper limb, 45.8% of the respondents answered that they could not tie a knot alone, and 25.4% of the participants said that they could not use chopsticks alone (Table 8). Furthermore, when asked about the time it took to get used to each ADL without the help of others, respondents answered that they got used to the activities in ≤6 months except for tying knots and using chopsticks: more than 38.8% and 26.5% responded that these respective tasks are almost impossible (Table 9). Therefore, there is an urgent need to develop a prosthetic arm capable of performing these activities. In particular, the use of chopsticks is included in the Korean-style daily life measurement tool; therefore, if a prosthetic hand that assists people in using chopsticks is developed, it would help improve self-reliance and the quality of life of people with disabilities.

## 5. Conclusions

This study contributes to the development of new prostheses by examining the status and use of the prosthesis, satisfaction with the existing prosthesis, and factors that satisfy/dissatisfy disabled individuals with upper limb amputation. This study indicates that the elbow-wrist was the most common amputation that the respondents underwent, and most of the respondents (98.3%) had acquired disabilities due to accidents. Next, 70.7% of disabled individuals with upper limb amputation in Korea use several cosmetic prostheses when the degree of disability is not severe. On average, people used cosmetic prosthesis for 21 years for 11.92 h a day.

Respondents who have used a prosthesis but do not currently use one cited discomfort (wearing, weight, discomfort in detachment) and dissatisfaction with the design and function as the main reasons for discontinuing the use of a prosthesis. In the case of the unilateral amputees, they responded that they did not need a prosthetic hand due to a complementing healthy upper limb. Participants responded that the price of cosmetic prosthesis was low; however, that they were dissatisfied with the esthetic and functional aspects of the assistive device. On the other hand, electronic prostheses are expensive, so accessibility is low, and it is made of mechanical elements, resulting in poor esthetics. In addition, 50% of the prosthesis users responded that they were not yet accustomed to the prostheses.

Despite continuous research by many countries and institutions, many disabled individuals with upper limb amputation are not satisfied with the function and design of the prosthetic upper limbs, and only use prostheses when necessary. Sokolowski [38] pointed out that because users interface closely with the products they use, understanding directly from users why they need them and their performance expectations is essential to discovering information that product, technological, or market cannot convey. This study proposed the development of a prosthesis that fulfills a user’s needs by clarifying the status of the prosthesis used by the upper limb amputee and the degree of satisfaction with the prosthesis currently being used. As a result, most Korean upper limb amputees use inexpensive cosmetic prostheses, and their preferred prosthetic color was skin color. According to Wijk and Carlsson [39], cosmetic hands are designed for esthetic purposes only, so it is important to implement shapes and colors that look like actual hands. However, according to this study, patients with upper limb amputation responded that when choosing a cosmetic hand, the ease of attachment and detachment, weight of the prosthesis, and its fit were more important than its color and design. The results of this study show that it is more important to develop a prosthesis that is easy to attach and detach, is lightweight, and comfortable to wear than other attributes such as color, shape, and design, even when developing a cosmetic prosthesis. In addition, respondents with a long cut due to an elbow amputation responded that a prosthetic hand made to fit the length of their hand was too heavy to regularly wear and was uncomfortable to use for daily tasks. Respondents indicated that they needed a variety of prosthetics designed according to the purpose and situation, such as for sports (golf, cycling, etc.) and everyday use.

Finally, in examining the degree of ADL performance of the upper limb amputee, most of the respondents answered that buttoning shirts, tying knots, and using chopsticks were difficult or impossible to do on their own. Therefore, prostheses that can assist people in performing these tasks can greatly improve the quality of amputees’ lives.

This study aimed to help develop a user-centered prosthetic arm that can help amputees’ daily lives by decreasing the psychological and social loss people experience due to upper limb amputation. If various prostheses are developed based on the results of this study, new prostheses that can satisfy both the needs and demands of users can be developed. Furthermore, some assistive technologies are classified as medical devices and are, therefore, covered by the Medical Devices Directive. This imposes higher criteria on the devices, which may lead to improvements but also makes it more expensive and time consuming to reach the market [40]. In the future, more disability-friendly regulation environment will also be needed to activate user-centered prosthetic development that can implement various functions and purposes and make it easier for consumers to purchase.

Although the study focused on Korea, it is not reasonable to generalize its results to amputees in the country. First, because only upper limb prosthetic users were included and second, because the statistical representativeness of the subjects was not intended, in the present study. In the future, if large-scale surveys with amputees of various ages and occupations, and in-depth interviews with amputees and experts are conducted, it will be able to contribute to the acceptance of amputee’s prostheses and improvement of their quality of life.

## Figures and Tables

**Table 1 healthcare-09-00839-t001:** General characteristics of the respondents (*n* = 59).

Demographic Characteristics		Number (%)	Value
Sex	Male	52 (88.1)	
	Female	7 (11.9)	
Age group at the time of amputation (years) *	10’s	1 (1.8)	59.04 ± 15.27
	30’s	5 (8.9)	
	40’s	9 (16.1)	
	50’s	12 (21.4)	
	60’s	12 (21.4)	
	70’s	13 (23.2)	
	80’s	4 (7.1)	
Cause of amputation	Trauma	53 (89.8)	
	Disease	3 (5.1)	
	The Korean war	2 (3.4)	
	Congenital	1 (1.7)	
Amputation side	Right	29 (49.2)	
	Left	20 (33.9)	
	Bilateral	10 (16.9)	
Level of amputation	Shoulder disarticulation	6 (8.8)	
	Transhumeral amputation	11 (16.2)	
	Elbow disarticulation	4 (5.9)	
	Transradial amputation	22 (32.4)	
	Wrist disarticulation	13 (19.1)	
	Partial hand and fingers amputation	12 (17.7)	
Period of prosthesis (years) *	≤1 year	7 (9.8)	
	1–5 years	8 (17.7)	
	5–10 years	8 (15.6)	
	10–20 years	6 (11.8)	
	20–30 years	6 (13.7)	
	30–40 years	4 (7.9)	
	40–50 years	8 (15.7)	
	≥50 years	4 (7.8)	
Amputation age (years) *			* 29.67 ± 14.41
Time since amputation (years) *			* 30.36 ± 19.01

* Values are expressed as the mean ± standard deviation.

**Table 2 healthcare-09-00839-t002:** Experience of prosthesis use (*n* = 59).

Experience of Prosthesis Use		Number (%)
Experience of prosthesis use	Used	55 (93.2)
	Never used	4 (6.8)
Prosthesis type experienced (multiple response)	Cosmetic hands	48 (56.5)
	Cable-activated prosthesis (Hook)	22 (25.9)
	Cable-activated prosthesis (Hands)	9 (10.6)
	Electric-powered prosthetic hand	6 (7.0)
Prosthetic upper limb type presently used	Cosmetic hands	41 (70.7)
	Cable-activated prosthesis (Hook)	12 (20.7)
	Cable-activated prosthesis (Hands)	3 (5.2)
	Electric-powered prosthetic hand	2 (3.4)
Prosthetic upper limb color used	Skin	50 (96.2)
	Black	1 (1.9)
	Grey	1 (1.9)

**Table 3 healthcare-09-00839-t003:** Discontinuance of prosthesis.

Discontinuance of Prosthesis Use		Number (%)
Use of prosthetic upper limb	Use	49 (89.1)
	Stopped use	6 (10.9)
Reason for discontinuance of prosthetic upper limb (multiple response)	Design dissatisfaction	7 (21.9)
	Fit dissatisfaction	5 (15.6)
	Weight dissatisfaction	5 (15.6)
	Function dissatisfaction	4 (12.5)
	Inconvenient attachment and detachment	4 (12.5)
	Unnecessary	3 (9.3)
	Price dissatisfaction	2 (6.3)
	Medical reasons	2 (6.3)

**Table 4 healthcare-09-00839-t004:** Rehabilitation training experience for prosthetic upper limb use.

Rehabilitation Training Experience for Prosthetic Upper Limb Use		Number (%)
Rehabilitation training experience	Yes	16 (29.1)
	No	39 (70.9)
Location for rehabilitation training	Hospital	14 (87.5)
	Prosthetic upper limb manufacturer	2 (12.5)

**Table 5 healthcare-09-00839-t005:** Usage status of the prosthetic upper limb (*n* = 49).

Usage Status of the Prosthesis		Number (%)	Value
Daily wear time of the prosthesis (hours) *	≤6 h	6 (12.2)	12.06 ± 4.785
	6–12 h	13 (26.6)	
	12–18 h	25 (51.0)	
	≥18 h	5 (10.2)	
Time required for getting accustomed to the prosthesis	≤6 months	8 (16.3)	
	6 months–1 year	5 (10.2)	
	1–3 years	9 (18.4)	
	≥3 years	3 (6.1)	
	Not comfortable	24 (49.0)	

* Values are expressed as the mean ± standard deviation.

**Table 6 healthcare-09-00839-t006:** Satisfaction with the prosthetic upper limb (*n* = 55).

Factor	Mean	SD
Color	3.42	1.197
Design	3.57	1.667
Price	3.1	1.418
Weight	3.02	1.205
Function	3.02	1.147
Fit	3.2	1.268
Convenience in attachment and detachment	3.37	1.248

Note. Prosthetic upper limb non-users were excluded from the analysis. SD, standard deviation.

**Table 7 healthcare-09-00839-t007:** Consumer’s priorities for cosmetic hand and electric-powered prosthetic hand (*n* = 55).

Attributes	Cosmetic Hand		Electric-Powered Prosthetic Hand		*t*-Value	*p*-Value
	Mean	SD	Mean	SD		
Color	4.38	1.114	4.35	1.160	0.442	0.661
Design	4.46	0.989	4.43	1.015	0.329	0.744
Price	4.27	1.217	4.43	1.168	−1.640	0.110
Weight	4.56	0.998	4.72	0.513	−1.234	0.226
Function	4.26	1.146	4.74	0.443	−2.843 **	0.008
Fit	4.51	0.818	4.71	0.519	−1.484	0.147
Convenience in attachment and detachment	4.63	0.598	4.66	0.591	−0.373	0.711

** *p* < 0.001. SD, standard deviation.

**Table 8 healthcare-09-00839-t008:** ADL performance assessment of upper-limb amputees (*n* = 59).

Degree of Dependencies	Washing	Grooming	Dressing			Eating		
			Buttoning Shirts	Closing the Zipper of Pants	Tying Shoelaces	Using a Spoon	Using a Fork	Using Chopsticks
Impossible alone	6.8% (*n* = 4)	5.1% (*n* = 3)	11.9% (*n* = 7)	8.5% (*n* = 5)	32.2% (*n* = 19)	1.7% (*n* = 1)	1.7% (*n* = 1)	16.9% (*n* = 10)
Very difficult alone	6.8% (*n* = 4)	11.9% (*n* = 7)	13.6% (*n* = 8)	8.5% (*n* = 5)	13.6% (*n* = 8)	5.1% (*n* = 3)	5.1% (*n* = 3)	8.5% (*n* = 5)
Need a little help	25.4% (*n* = 15)	23.7% (*n* = 14)	10.2% (*n* = 6)	10.2% (*n* = 6)	23.7% (*n* = 14)	15.3% (*n* = 9)	15.3% (*n* = 9)	6.8% (*n* = 4)
Possible alone	61.0% (*n* = 36)	59.3% (*n* = 35)	64.4% (*n* = 38)	72.9% (*n* = 43)	30.5% (*n* = 18)	78.0% (*n* = 46)	78.0% (*n* = 46)	67.8% (*n* = 40)
**Degree of** **dependencies**	**Toileting**	**Using a Cell Phone**	**Using a Remote Controller**	**Using a Computer Mouse**	**Using a Keyboard**			
Impossible alone		5.1% (*n* = 3)	1.7% (*n* = 1)	3.4% (*n* = 2)	5.1% (*n* = 3)			
Very difficult alone	5.1% (*n* = 3)	3.4% (*n* = 2)	1.7% (*n* = 1)	5.1% (*n* = 3)	3.4% (*n* = 2)			
Need a little help	15.3% (*n* = 9)	11.9% (*n* = 7)	11.9% (*n* = 7)	11.9% (*n* = 7)	23.7% (*n* = 14)			
Possible alone	79.7% (*n* = 47)	79.7% (*n* = 47)	84.7% (*n* = 50)	79.7% (*n* = 47)	67.8% (*n* = 40)			

Note. Prosthetic upper limb non-users were excluded from the analysis. ADL, activities of daily living.

**Table 9 healthcare-09-00839-t009:** Time required for ADL performance using prosthetic upper limb (*n* = 49).

Time Required for ADL Performance	Washing	Grooming	Dressing			Eating		
			Buttoning Shirts	Closing the Zipper of Pants	Tying Shoe-Laces	Using a Spoon	Using a Fork	Using Chopsticks
≥1 week	24.5% (*n* = 12)	20.4% (*n* = 10)	20.4% (*n* = 10)	22.4% (*n* = 11)	10.2% (*n* = 5)	34.7% (*n* = 17)	36.7% (*n* = 18)	28.6% (*n* = 14)
1 week–3 months	20.4% (*n* = 10)	16.3% (*n* = 8)	14.3% (*n* = 7)	18.4% (*n* = 9)	12.2% (*n* = 6)	18.4% (*n* = 9)	18.4% (*n* = 9)	16.3% (*n* = 8)
3–6 months	18.4% (*n* = 9)	18.4% (*n* = 9)	24.5% (*n* = 12)	22.4% (*n* = 11)	12.2% (*n* = 6)	12.2% (*n* = 6)	12.2% (*n* = 6)	10.2% (*n* = 5)
6 months–1 year	8.2% (*n* = 4)	14.3% (*n* = 7)	14.3% (*n* = 7)	16.3% (*n* = 8)	10.2% (*n* = 5)	12.2% (*n* = 6)	10.2% (*n* = 5)	10.2% (*n* = 5)
1–2 year	8.2% (*n* = 4)	8.2% (*n* = 4)	4.1% (*n* = 2)	8.2% (*n* = 4)	8.2% (*n* = 4)	6.1% (*n* = 3)	8.2% (*n* = 4)	6.1% (*n* = 3)
2–3 year	4.1% (*n* = 2)	8.2% (*n* = 4)	6.1% (*n* = 3)	2.0% (*n* = 1)	8.2% (*n* = 4)	8.2% (*n* = 4)	6.1% (*n* = 3)	2.0% (*n* = 1)
Impossible	16.3% (*n* = 8)	14.3% (*n* = 7)	16.3% (*n* = 11)	10.2% (*n* = 5)	38.8% (*n* = 19)	8.2% (*n* = 4)	8.2% (*n* = 4)	26.5% (*n* = 13)
**Time required for ADL performance**	**Toileting**	**Using a Cell Phone**	**Using Remote Controller**	**Using Computer Mouse**	**Using Keyboard**			
≥1 week	32.7% (*n* = 16)	38.8% (*n* = 19)	44.9% (*n* = 22)	36.7% (*n* = 18)	32.7% (*n* = 16)			
1 week–3 months	24.5% (*n* = 12)	16.3% (*n* = 8)	14.3% (*n* = 7)	16.3% (*n* = 8)	14.3% (*n* = 7)			
3–6 months	16.3% (*n* = 8)	18.4% (*n* = 9)	16.3% (*n* = 8)	16.3% (*n* = 8)	18.4% (*n* = 9)			
6 months–1 year	0% (*n* = 0)	2.0% (*n* = 1)	2.0% (*n* = 1)	4.1% (*n* = 2)	4.1% (*n* = 2)			
1–2 year	6.1% (*n* = 3)	4.1% (*n* = 2)	6.1% (*n* = 3)	4.1% (*n* = 2)	4.1% (*n* = 2)			
2–3 year	6.1% (*n* = 3)	6.1% (*n* = 3)	6.1% (*n* = 3)	8.2% (*n* = 4)	10.2% (*n* = 5)			
Impossible	14.3% (*n* = 7)	14.3% (*n* = 7)	10.2% (*n* = 5)	14.3% (*n* = 7)	16.3% (*n* = 8)			

Note. Only current prosthetic upper limb users were included in the analysis. ADL, activities of daily living.

## Data Availability

The data presented in this study are available on request from the corresponding author. The data are not publicly available due to privacy.

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
