# Peer review of "A User-Driven Approach to Prosthetic Upper Limb Development in Korea"

_healthcare, 2021, doi:10.3390/healthcare9070839_

Round 1
Reviewer 1 Report
Analysis user needs for customized prosthetic limb development in Korea
Thanks for the opportunity to review this interesting study seeking to understand the scale of prosthetic usage in Korea and the reasons for compliance or non compliance with use.
Strengths:
Large sample size, data collected across a variety of issues pertaining to compliance with prosthesis use.
Weaknesses:
Overall, the content of the paper is a valuable contribution but I have some concerns about the current structure of the study. The research questions proposed in the introduction have not been adequately linked to the methods described. How do the questions used in the survey answer these research questions – which relate to which research question and how is the survey validated to ensure that this occurs? The methods section needs major revision to provide reviewers and readers with enough information that they can fully interpret the results.
Introduction
Major comments
The introduction presents a very thorough literature review into different aspects of prosthetic usage. The authors set up a set of research questions based on the review. I suggest reducing the length of the literature review and instead focussing the introduction on the rational of the study and aims based on what is missing from the current evidence.
Line 44- line 48 talks about inducing people to understand… – this is quite a strong statement, perhaps the authors mean something like ;it is necessary to develop technology to facilitate participation’ rather then placing the onus on the prosthetic user?
Methods
This section does not include a copy of the full questionnaire or reference to supplementary material where this is placed. It is therefore not currently possible to fully review the methods. This should be included in the questionnaire composition section
References for each of the surveys mentioned, where questions were selectively used, should also be included in this section.
Was any validation of this new survey performed? How do the authors know that the responses are going to answer their research questions?
Results
This section is nicely laid out according to each research question but there is not enough information about eh survey presented for me to properly review the results. Authors need to clarify either in the methods or results what the questions that were asked were eg: what is ‘use’ and what is ‘disuse’?, what is difference between ‘the prosthesis they used’ and the ‘prosthesis currently in use’. This section can be reviewed more thoroughly once this change is made.
Conclusions
First sentence and final paragraph states that this study contributes to the development of new prostheses, but the survey seeks to understand reasons for use and disuse, not develop prostheses. Suggest clarifying.
Minor comments
Title needs to be reviewed for grammar, also suggest including ‘upper limb’ as this study is specific to those.
This section needs to be reviewed for grammatical errors.
‘lower amputation’ and ‘upper amputation’ should read lower limb amputation and upper limb amputation.
Suggest changing phrase ‘must amputate a body part’ to ‘undergo amputation’
What is an amputation disability?
What is a ‘severe amputee’?
Methods
The terms disability and amputation have been used interchangeably, the authors should be specific.
Table 1
Age- this is by decade, not a continuous variable
Typo ‘Case’ should be ‘cause’
Korean war is very specific – does this not fall under trauma?
Amputation side – ‘both’ should be replaced with ‘bilateral’
Amputation age – consider if ‘age at time of amputation’ is clearer
Duration of amputation – consider if ‘time since amputation’ is clearer
Reviewer 2 Report
Overview and general recommendations:
The paper focus on the need to assess the perceptions and needs of people with upper-limb amputation in relation to the use of prosthetics, to perform activities for daily living. The methodology used was a set of questionnaires.
Results of the paper can be important and add knowledge to the current framing of the development of prosthesis in Korea.
The main strength of the paper is the focus on an especially important and sometime neglected topic. However, the study presents several important flaws that need to be addressed, namely the need to update literature review (studies from 1981 to 1988 on topics that are recently studied are inadmissible) and the justification of the methods. For this reason, I did not focused on the results of the study, since its premisses were not clear, in my perspective.
Since the suggested revision will imply more than a mere language check (also much needed!) or small complements, I recommend that a major revision is necessary. However, I strongly encourage the authors to consider my suggestions, complement the paper and re-submit it.
Below, in more detail, I explain my perspective and concerns.
Major comments:
- The paper would really benefit from a thorough professional language check.
- Abstract: a reference to the methodology is missing.
- The references that support a research paper are of great importance, as we all know. I wonder why the authors based their work on references that, considering the topic of the paper, are clearly outdate. The literature review news to be reviewed, considering recent publications!
In addition, the internet link of reference [1] does not work and was last accessed by the authors in 2018 (!).
The authors need/must make a literature review based on recent publications on the topic.
- The terminology used in the paper when it comes to people who have been submitted to amputation, needs/must be revised. (“physically disabled”, “handicapped people”)
The same applies for some paragraphs that, in my opinion, are offensive and bias (see lines 148-149 for instance).
- As the title of the paper mentions, the authors focus on “customized prosthetics”. I would vividly suggest the authors to rethink their approach. I think that the aim of the paper is in fact to consider a User-centred design, to ensure the inclusion of the user at all levels of the technology development in a co-creation, or co-design approach. Or by considering a universal design approach (not necessarily have its original meaning to one fits “all”), and not on having “customized” prosthetics.
Results to a study conducted for the European Parliament on the perceptions and needs of people with disabilities could be interesting for the authors. Although prosthesis was not focus, the general results are nevertheless important.
To the study: Assistive technologies for people with disabilities - Think Tank (europa.eu)
- The reasons for performing the study are not clearly identified in the introduction. The authors should identify the existing problem and how this paper can contribute to resolve or assist in the problem.
- Lines 44 – 48: the authors suggest that depression and social isolation could be overcome by “physically disabled to properly understand the importance and necessity of physical activity and actively participate”. Is this really the focus? Participate in what? Will really the development of a technology resolve isolation problems? This paragraph needs to be better explained.
- Since the study focus on Korea, a short introduction to Korean’s health system could be provided to guide the reader concerning accessibility and reimbursement issues of assistive technologies.
- To better guide the reader and have a better flow in the paper, all research questions should be identified, for instance at the end of the literature review.
In addition, if the authors want to keep them as research questions they need to be reformulated as indeed research questions. For example, 1: What is the current status of the of prosthetics… and not Investigate the current status of the use of prosthetic.
- Line 221-223: This information should be in section 3. Methods
- In the methods, it is mentioned that participants of the study are “athletes” (line 192) and people going into a rehabilitation process. No information is provided on how the authors selected from this groups, people that were “suffering from a significant psychological and social (interaction) loss due to upper amputation” (line 62-65).
In addition, athletes that use prosthesis for sports, have special requirements compared to prosthesis that are used to conduct normal routine tasks. Why this specific recruitment took place, if the focus is on prothesis that can assist in ADL?
- Line 208 and 212-213: the authors mentioned that existing questionnaires were adapted to serve the purpose of the study. However, no information is given on what was changed and why and how the validation of the adapted questionnaires was conducted. Also it is not clear if the questionnaires were self-administrated or if the researcher conducted the questions.
- The data from when to when the data was collected is missing.
- Line 291-293: this information should be in Section 3. Methods.
- Since only 59 responses, collected in two institutions, were consider valid for analysis, how do the authors justify the generalization of results to Korea?
Minor comments:
- Please consider include additional keywords in the abstract, such as for instance: user’s needs, user’s requirements, Assistive Technology, technology development
- Line 18: Since these are the results of the study, the sentence could start my mentioning: “Results of the study showed that” the main reasons (…)
- Line 37: concerning the incident rates, a reference should be provided.
- Lines 53-54: this paragraph needs to be developed and explain the reasons why the assistive devices were not purchased.
- Line 53: the information on what ref [1] refers to, should be provide already in line 34-36.
- Line 57- 60: reference(s) should be provided.
- Line 86-88: a reference should be provided.
- Line 101 – 119: this information could be provided using the authors own words to summarize the findings. It seems that the questions in the study were made with multiple choice answers, and thus no need to have the questions replicated in the article.
- Line 139: Please check for consistency of concepts: “auxiliary equipment” should be replaced by “assistive technologies”.
- Line 390: Information on the Funding is missing
Round 2
Reviewer 2 Report
Dear Authors,
Thank you for considering my comments and suggestions. The paper has improved, and the exposition of the research is now clearer.
I still have few comments that I would like to ask you to reflect and consider. For this reason, my decision to accept the paper, after minor revisions. Again, I encourage the authors to address them a resubmit the paper, due to its importance in the prosthetic development field and the need from developers to be opened to co-creation or co-development.
Comments:
Response 2: methodology reference in the abstract.
I would suggest making a more objective approach:
A modified questionnaire was conducted to upper limb prosthetic users and results were analysed using descriptive statistics.
Response 8:
[Revised] Therefore, to reduce depression and relieve social isolation in people with disabilities, it is necessary to develop assistive devices that enable physical activity.
I still struggle with this sentence.
The authors assume that technology can solve isolation problems by enabling people to do physical activity. So, what happens to those who do not want a prosthesis?
Assistive technology can play a role, but one cannot assume that it is “necessary to develop assistive devices that enable physical activity” to reduce depression and relieve social isolation.
I suggest a change in the sentence to:
Therefore, to reduce depression and relieve social isolation in people with disabilities, the develop of assistive devices that enable physical activity, can play a role.
Response 10:
[Revised] Therefore, this study created the following research questions to help in the de-velopment of customized prosthetics that can assist the active physical activity of the disabled with upper limb amputation and assist them in daily life movements.
Investigate the actual use of prostheses of upper limb amputees.
Research Question 2. Explain the reasons for prosthesis rejection by upper limb amputees.
Research Question 3. Explain the factors upper limb amputees consider when choosing a prosthetic arm.
Research Question 4. Identify the degree of daily life movements of the disabled with upper limb amputation.
Research Question 5. Investigate daily life movements that are difficult for the disabled with a prosthetic limb amputation to perform.
I advise the authors to read, for instance Bryman, Alan. 2012. Social Research Methods. 4th ed. New York: Oxford University Press, Inc.
A research question is indeed a question! And as such, should be formulated with a question mark at the end. As it is presented, the 5 sentences are aims and not research questions. I strongly advise the authors to review them.
Response 13
Although the authors mention that the questionnaire was adapted and clearly identify its sources, issues of validity of the adapted questionnaire are still not presented. it is not sufficient to mentioned that the validity was made by experts. It would be beneficial, if the authors, even if briefly, present the process of validity, meaning a reference to experts' comments and how the authors addressed them should be presented, to make the research more sound and more meaningful, also considering that no reliability test was conducted (I am assuming that).
For this last point, reliability, authors should present Cronbach's alpha value.
Point 14: Line 291-293: this information should be in Section 3. Methods.
Response 14: Should we move line 291-293 to methods section?
Yes! The information concerning the use of a Likert scale should be provided in the methods section, when presenting the questionnaire (that by the way should be included in the article as appendix or should be made available by the authors).
In the description of the questionnaire, it should be mentioned if open or closed questions were use, and if scales were used in the closed questions, providing the scales designation.
Point 15: Since only 59 responses, collected in two institutions, were consider valid for analysis, how do the authors justify the generalization of results to Korea?
Response 15: We added to the conclusion that the inability to generalize the results of this study is a limitation of this study.
[Revised] This study is only for amputees in Korea, and it is difficult to generalize the results of this study. In the future, if large-scale surveys with amputees of various ages and occupations, and in-depth interviews with amputees and experts are conducted, it will be able to contribute to the acceptance of amputee's prostheses and improvement of their quality of life.
I would suggest a different more concrete approach to the identified “limitation”:
Although the study focused on Korea, it is not reasonable to generalize its results to amputees in the country. First, because only upper limb prosthetic users were included and second because the statistical representativeness of the subjects was not intended, in the present study.
Extra comment:
In line 58 of the paper new version, the authors say:
“However, recent research has focused on the development of new prosthetics utilizing various functions and the latest technologies, investigations of satisfaction with the prostheses and their rate of use have been limited [8, 9, 10].”
It is a bit strange that in a study that refers to limb prostheses, a reference to a study that focus on dental prostheses is presented. Maybe the authors should check if the correct paper is being referred.
Author Response
Thank you for your valuable comments. Please see the attached response note.

Round 3
Reviewer 2 Report
Dear Authors,
Thank you for considering my comments and suggestions, in this second round.
My still last comment to you, and as a suggestion for future papers is to provide for a short paragraph regarding the expert’s feedback to the modified questionnaire, if you do it again. This increases the soundness of your work and provides for more transparency.
Looking forward to see your paper published and congratulations on your study and work.
Author Response
Dear reviewer,
Thank you for taking your valuable time to comment on our manuscript.
Questions that have been corrected or added after receiving expert feedback have been explained in the text. It was also stated in the text that the questionnaire was evaluated by the expert group and IRB members as appropriate for conducting the study.
For the study, researchers first constructed a draft questionnaire by referring to previous studies. Afterward, this questionnaire was evaluated by a group of experts, including professors of rehabilitation medicine, prosthetic developers, and employees of prosthetic manufacturers, and was finally used for research after reviewing two persons with disabilities with upper limb amputation using prosthetic prostheses. As a result of the review, the overall length of the questionnaire was reduced in consideration of the condition of the respondent, and only those who wanted to respond were allowed to respond to sensitive responses such as educational background and marital status. In the case of the use of the prosthesis, questions were added to answer the reasons why they did not continue to use the prosthesis because the pilot study had subjects who stopped using the prosthesis. For the question about the time it takes for the prosthesis to be convenient to use, the response 'It is not convenient to use the prosthesis' was added to reflect the opinion that the prosthesis is not convenient to use even after a long period of time. The revised questionnaire was conducted after receiving final feedback from experts and IRB members that it was appropriate for conducting the study.